# Backstepping Temporal Difference Learning

**Han-Dong Lim**
Department of Electrical Engineering
KAIST, Daejeon, 34141, South Korea
limaries30@kaist.ac.kr

**Donghwan Lee**
Department of Electrical Engineering
KAIST, Daejeon, 34141, South Korea
donghwan@kaist.ac.kr

## Abstract

Off-policy learning ability is an important feature of reinforcement learning (RL) for practical applications. However, even one of the most elementary RL algorithms, temporal-difference (TD) learning, is known to suffer form divergence issue when the off-policy scheme is used together with linear function approximation. To overcome the divergent behavior, several off-policy TD-learning algorithms, including gradient-TD learning (GTD), and TD-learning with correction (TDC), have been developed until now. In this work, we provide a unified view of such algorithms from a purely control-theoretic perspective, and propose a new convergent algorithm. Our method relies on the backstepping technique, which is widely used in nonlinear control theory. Finally, convergence of the proposed algorithm is experimentally verified in environments where the standard TD-learning is known to be unstable.

## 1 Introduction

Since Mnih et al. (2015), which has demonstrated that deep reinforcement learning (RL) outperforms human in several video games (Atari 2600 games), significant advances has been made in RL theory and algorithms. For instance, Van Hasselt et al. (2016); Lan et al. (2020); Chen et al. (2021) proposed some variants of the so-called deep Q-network (Mnih et al., 2015) that achieves higher scores in Atari games than the original deep Q-network. An improved deep RL was developed in Badia et al. (2020) that performs better than average human scores across 57 Atari games. Not only performing well in video games, but Schrittwieser et al. (2020) also have shown that an RL agent can self-learn chess, Go, and Shogi. Furthermore, RL has shown great success in real world applications, e.g., robotics (Kober et al., 2013), healthcare (Gottesman et al., 2019), and recommendation systems (Chen et al., 2019).

Despite the practical success of deep RL, there is still a gap between theory and practice. One of the notorious phenomena is the deadly triad (Sutton & Barto, 2018), the diverging issue of the algorithm when function approximation, off-policy learning, and bootstrapping are used together. One of the most fundamental algorithms, the so-called temporal-difference (TD) learning (Sutton, 1988), is known to diverge under the deadly triad, and several works have tried to fix this issue for decades. In particular, the seminar works Sutton et al. (2008; 2009) introduced the so-called GTD, gradient-TD2 (GTD2), and TDC, which are off-policy, and have been proved to be convergent with linear function approximation. More recently, Ghiassian et al. (2020) suggested regularized version of TDC called TD learning with regularized correction (TDRC), and showed its favorable features under off-policy settings. Moreover, Lee et al. (2021) developed several variants of GTD based on primal dual formulation.

On the other hand, backstepping control (Khalil, 2015) is a popular method in designing stable controllers for nonlinear systems with special structures. The design technique offers a wide range of stable controllers, and is proved to be robust under various settings. It has been used in various fields including quadrotor helicopters (Madani & Benallegue, 2006), mobile robots (Fierro & Lewis, 1997), and ship control (Fossen & Strand, 1999). Using backstepping control technique, in this paper, we develop a new convergent off-policy TD-learning which is a single time-scale algorithm.

In particular, the goal of this paper is to introduce a new unifying framework to design off-policy TD-learning algorithms under linear function approximation. The main contributions are summarized as follows:

- We propose a systemic way to generate off-policy TD-learning algorithms including GTD2 and TDC from control theoretic perspective.

- Using our framework, we derive a new TD-learning algorithm, which we call backstepping TD (BTD).

- We experimentally verify its convergence and performance under various settings including where off-policy TD has known to be unstable.

In particular, most of the previous works on off-policy TD-learning algorithms (e.g., GTD2 and TDC) are derived based on optimization perspectives starting with an objective function. Then, the convergence is proved by proving stability of the corresponding O.D.E. models. In this paper, we follow reversed steps, and reveal that an off-policy TD-learning algorithm (called backstepping TD) can be derived based on control theoretic motivations. In particular, we develop stable O.D.E. models first using the backstepping technique, and then recover back the corresponding off-policy TD-learning algorithms. The new analysis reveals connections between off-policy TD-learning and notions in control theory, and provides additional insights on off-policy TD-learning with simple concepts in control theory. This sound theoretical foundation established in this paper can potentially motivate further analysis and developments of new algorithms.

Finally, we briefly summarize TD learning algorithms that guarantee convergence under linear function approximation. GTD (Sutton et al., 2008), GTD2 and TDC (Sutton et al., 2009) have been developed to approximate gradient on mean squared projected Belllman error. Later, GTD and GTD2 has been discovered to solve minimax optimization problem (Macua et al., 2014; Liu et al., 2020). Such sadde-point view point of GTD has led to many interesting results including Du et al. (2017); Dai et al. (2018); Lee et al. (2021). TDRC (Ghiassian et al., 2020) adds an additional term similar to regularization term to one-side of parameter update, and tries to balance between the performance of TD and stability of TDC. TDC++ (Ghiassian et al., 2020) also adds an additional regularization term on both sides of the parameter update. Even though TDRC shows good performance, it uses additional parameter condition to ensure convergence, whereas TDC++ does not.

## 2 PRELIMINARIES

### 2.1 NONLINEAR SYSTEM THEORY

Nonlinear system theory will play an important role throughout this paper. Here, we briefly review basics of nonlinear systems. Let us consider the continuous-time nonlinear system

$$\dot{x}_t = f(x_t, u_t), \quad x_0 \in \mathbb{R}^n, \tag{1}$$

where $x_0 \in \mathbb{R}^n$ is the initial state, $t \in \mathbb{R}, t \geq 0$ is the time, $x_t \in \mathbb{R}^n$ is the state, $u_t \in \mathbb{R}^n$ is the control input, and $f : \mathbb{R}^n \times \mathbb{R}^n \to \mathbb{R}^n$ is a nonlinear mapping. An important concept in dealing with nonlinear systems is the equilibrium point. Considering the state-feedback law $u_t = \mu(x_t)$, the system can be written as $\dot{x}_t = f(x_t, u_t) = f(x_t, \mu(x_t)) =: f(x_t)$, and a point $x = x^e$ in the state-space is said to be an equilibrium point of (1) if it has the property that whenever the state of the system starts at $x^e$, it will remain at $x^e$ (Khalil, 2015). For $\dot{x}_t = f(x_t)$, the equilibrium points are the real roots of the equation $f(x) = 0$. The equilibrium point $x^e$ is said to be globally asymptotically stable if for any initial state $x_0 \in \mathbb{R}^n$, $x_t \to x^e$ as $t \to \infty$.

An important control design problem is to construct a state-feedback law $u_t = \mu(x_t)$ such that the origin becomes the globally asymptotically stable equilibrium point of (1). To design a state-feedback law to meet such a goal, control Lyapunov function plays a central role, which is defined in the following definition.

**Definition 2.1** (Control Lyapunov function (Sontag, 2013)). *A positive definite function $V : \mathbb{R}^n \to \mathbb{R}$ is called a control Lyapunov function (CLF) if for all $x \neq 0$, there exists a corresponding control input $u \in \mathbb{R}^m$ that satisfies the inequality, $\nabla_x V(x)^\top f(x, u) < 0$ for all $x \neq 0$.*

Once such a CLF is found, then it guarantees that there exists the control law that stabilizes the system. Moreover, the corresponding state-feedback control law can be extracted from the CLF, e.g., $\mu(x) = \arg\min_u \nabla_x V(x)^\top f(x, u)$ provided that the minimum exists and unique. The concept of control Lyapunov function will be used in the derivations of our main results. For the autonomous

system, $\dot{x}_t = f(x_t)$, and Lypaunov function $V : \mathbb{R}^n \to \mathbb{R}$, Lie derivative is defined as $\mathcal{L}_f V(x) := \nabla_x V(x)^\top f(x)$ so that $\dot{V}(x_t) = \mathcal{L}_f V(x_t)$ along the solution.

## 2.2 STOCHASTIC APPROXIMATION AND O.D.E. APPROACH

Including Q-learning (Watkins & Dayan, 1992) and TD-learning (Sutton, 1988), reinforcement learning algorithms can be considered as stochastic approximation (Robbins & Monro, 1951) described by

$$x_{k+1} = x_k + \alpha_k(f(x_k) + \epsilon_k), \tag{2}$$

where $f : \mathbb{R}^n \to \mathbb{R}^n$ is a nonlinear mapping, and $\epsilon_k$ is an i.i.d. noise. Borkar and Meyn theorem (Borkar & Meyn, 2000) is a well-known method to bridge the asymptotic convergence of stochastic approximation and the stability of its corresponding O.D.E. model, which can be expressed as

$$\dot{x}_t = f(x_t), \quad x_0 \in \mathbb{R}^n, \tag{3}$$

where $x_0 \in \mathbb{R}^n$ is initial state, and $t \in \mathbb{R}, t \geq 0$ is the time.

Borkar and Meyn theorem (Borkar & Meyn, 2000) states that under the conditions in Assumption 7.1 in the Appendix, global asymptotic stability of the O.D.E. (3) leads to asymptotic convergence of the stochastic approximation update (2), which is formally stated in the following lemma.

**Lemma 2.1** (Borkar and Meyn theorem (Borkar & Meyn, 2000)). *Suppose that Assumption 7.1 in the Appendix holds, and consider the stochastic approximation in (2). Then, for any initial $x_0 \in \mathbb{R}^n$, $\sup_{k \geq 0} \|x_k\| < \infty$ with probability one. In addition , $x_k \to x^e$ as $k \to \infty$ with probability one, where $x^e$ is the unique equilibrium point of the O.D.E. in (3).*

The main idea of Borkar and Meyn theorem is as follows: iterations of a stochastic recursive algorithm follow the solution of its corresponding O.D.E. in the limit when the step-size satisfies the so-called Robbins-Monro condition (Robbins & Monro, 1951) in (33) in the Appendix. Therefore, by proving asymptotic stability of the O.D.E., we can induce convergence of the original algorithm. In this paper, we will use an O.D.E. model of TD-learning, which is expressed as a linear time-invariant system.

## 2.3 BACKSTEPPING CONTROL

This section provides the concept of the backstepping control (Kokotovic, 1992; Khalil, 2015), which will be the main tool in this paper to derive TD-learning algorithms. The backstepping technique is a popular tool for generating a CLF (control Lyapunov function) for nonlinear systems with specific structures. In particular, let us start with the following general nonlinear system:

$$\dot{y}_t = f(y_t) + g(y_t)x_t \tag{4}$$
$$\dot{x}_t = u_t,$$

where $y_t \in \mathbb{R}^m, x_t \in \mathbb{R}^m$ are the states, $u_t \in \mathbb{R}^m$ is the input, and $f : \mathbb{R}^m \to \mathbb{R}^m$ and $g : \mathbb{R}^m \to \mathbb{R}$ are continuous functions. The first system is a nonlinear system with a particular affine structure, and the second system is simply an integrator. It can be seen as a cascade interconnection of two systems, where the second system's state is injected to the input of the first system. The backstepping control technique gives us a systematic way to generate a CLF for such particular nonlinear systems provided that the first system admits a CLF independently. To this end, we suppose that the first system admits a CLF. Through the backstepping approach, designing a stable control law for the above system can be summarized in the following steps:

Step 1. Consider $x_t$ in (4) as virtual input $\tilde{x}(y_t)$ (state-feedback controller), and consider the following system: $\dot{\lambda}_t = f(y_t) + g(y_t)\tilde{x}(y_t)$. Design $\tilde{x}(y_t)$ such that the above system admits a CLF $V$, i.e., it admits a positive definite and radially unbounded function $V$ such that its time derivative is negative definite, i.e.,$\dot{V}(y_t) < 0, \forall y_t \neq 0$.

Step 2. Denote the error between the virtual state-feedback controller $\tilde{x}(y_t)$ and state variable $x_t$ as $z_t := x_t - \tilde{x}(y_t)$. Now, rewrite the original O.D.E. in (4) with the new variable $(y_t, z_t)$:
$$\frac{d}{dt}\begin{bmatrix} y_t \\ z_t \end{bmatrix} = \begin{bmatrix} f(y_t) + g(y_t)\tilde{x}(y_t) + g(y_t)z_t \\ u_t - \dot{\tilde{x}}(y_t) \end{bmatrix}$$

Step 3. Design the control input $u_t$ such that the above system is stable. One popular choice is to consider the CLF $V_c(y_t, z_t) := V(y_t) + ||z_t||^2/2$, where $V(y_t)$ is defined in Step 1. Then choose $u_t$ such that the time derivative of $V_c(y_t, z_t)$ to be negative definite.

A simple example of designing stabilizing control law by backstepping technique is given in Appendix Section 7.3.

## 2.4 MARKOV DECISION PROCESS

In this paper, we consider a Markov decision process (MDP) characterized by the tuple $(\mathcal{S}, \mathcal{A}, \mathcal{P}, \gamma, r)$, where $\mathcal{S} := \{1, 2, \ldots, |\mathcal{S}|\}$ stands for the set of finite state space, $|\mathcal{S}|$ denotes the size of $\mathcal{S}$, $\mathcal{A} := \{1, 2, \ldots, |\mathcal{A}|\}$ denotes the set of finite action space, $|\mathcal{A}|$ is the size of $\mathcal{A}$, $\gamma \in (0, 1)$ is the discount factor, $\mathcal{P} : \mathcal{S} \times \mathcal{A} \times \mathcal{S} \to [0, 1]$ denotes the Markov transition kernel, and $r : \mathcal{S} \times \mathcal{A} \times \mathcal{S} \to \mathbb{R}$ means the reward function. In particular, if an agent at state $s \in \mathcal{S}$, takes action $a \in \mathcal{A}$, then the current state transits to the next state $s' \in \mathcal{S}$ with probability $\mathcal{P}(s, a, s')$, and the agent receives reward $r(s, a, s')$. Each element of the state to state transition matrix under policy $\pi$, denoted by $P^\pi \in \mathbb{R}^{|\mathcal{S}| \times |\mathcal{S}|}$ is $[P^\pi]_{ij} := \sum\limits_{a \in \mathcal{A}} \pi(a|i)\mathcal{P}(i, a, j), \quad 1 \leq i, j \leq |\mathcal{S}|$, where $[P^\pi]_{ij}$ corresponds to $i$-th row and $j$-th column element of matrix $P^\pi$. Moreover, the stationary state distribution induced by policy $\mu$, is denoted as $d^\mu : \mathcal{S} \to [0, 1]$, i.e., $d^{\mu\top} P^\mu = d^{\mu\top}$. With the above setup, we define the following matrix notations:

$$D^\mu := \begin{bmatrix} d^\mu(1) & & \\ & \ddots & \\ & & d^\mu(|\mathcal{S}|) \end{bmatrix} \in \mathbb{R}^{|\mathcal{S}| \times |\mathcal{S}|}, \quad R^\pi = \begin{bmatrix} \mathbb{E}_{a \sim \pi}[r(s, a, s')|s = 1] \\ \mathbb{E}_{a \sim \pi}[r(s, a, s')|s = 2] \\ \vdots \\ \mathbb{E}_{a \sim \pi}[r(s, a, s')|s = |\mathcal{S}|] \end{bmatrix} \in \mathbb{R}^{|\mathcal{S}|},$$

where $D^\mu$ is a diagonal matrix of the state distribution induced by behavior policy $\mu$, each element of $R^\pi$ is the expected reward under policy $\pi$ at the corresponding state. The policy evaluation problem aims to approximate the value function at state $s \in \mathcal{S}$, $v^\pi(s) := \mathbb{E}\left[\sum_{k=0}^\infty \gamma^k r(S_k, A_k, S_{k+1}) \middle| S_0 = s, \pi\right]$, where the trajectory is generated under policy $\pi : \mathcal{S} \times \mathcal{A} \to [0, 1]$. In this paper, we consider the linear function approximation to approximate the value function $v^\pi(s)$. In particular, we parameterize the value function $v^\pi(s)$ with $\phi^\top(s)\xi$, where $\phi : \mathcal{S} \to \mathbb{R}^n$ is a pre-selected feature vector with $\phi(s) := [\phi_1(s) \quad \cdots \quad \phi_n(s)], \phi_1, \ldots, \phi_n : \mathcal{S} \to \mathbb{R}$ are feature functions, and $\xi \in \mathbb{R}^n$ is the learning parameter. The goal of the policy evaluation problem is then to approximate the value function $v^\pi(s)$ using this linear parameterization, i.e., $\phi^\top(s)\xi \approx v^\pi(s)$. Moreover, using the matrix notation $\Phi := [\phi(1), \phi(2), \cdots, \phi(|\mathcal{S}|)]^\top \in \mathbb{R}^{|\mathcal{S}| \times n}$, called the feature matrix, the linear parameterization can be written in the vector form $\Phi\xi$. We also assume that $\Phi$ is full column rank matrix throughout the paper, which is a standard assumption (Sutton et al., 2008; 2009; Ghiassian et al., 2020; Lee et al., 2021).

## 2.5 TEMPORAL DIFFERENCE LEARNING

This section provides a brief background on TD-learning (Sutton, 1988). Suppose that we have access to stochastic samples of state $s_k$ from the state stationary distribution induced by the behavior policy $\mu$, i.e., $s_k \sim d^\mu(\cdot)$, and action is chosen under behavior policy $\mu$, i.e., $a_k \sim \mu(\cdot|s_k)$. Then, we observe the next state $s'_k$ following $s'_k \sim \mathcal{P}(\cdot, a_k, s_k)$, and receive the reward $r_k := r(s_k, a_k, s'_k)$. Using the simplified notations for the feature vectors $\phi_k := \phi(s_k), \quad \phi'_k = \phi(s'_k)$. the TD-learning update at time step $k$ with linear function approximation can be expressed as $\xi_{k+1} = \xi_k + \alpha_k \rho_k \delta_k(\xi_k)\phi_k$, where $\alpha_k > 0$ is the step-size, $\delta_k(\xi_k) := r_k + \gamma\phi_k'^\top\xi_k - \phi_k^\top\xi_k$ is called the temporal difference or temporal difference error (TD-error), and $\rho_k := \rho(s_k, a_k) = \frac{\pi(a_k|s_k)}{\mu(a_k|s_k)}$ is called the importance sampling ratio (Precup et al., 2001). The importance sampling ratio re-weights the TD-error to handle the mismatch between the behavior policy $\mu$ and target policy $\pi$. It is known that TD-learning with linear function approximation and off-policy learning scheme does not guarantee convergence in general. The above stochastic approximation aims to find fixed point of the following projected Bellman equation, which is, after some manipulations, expressed as:

$$\Phi^\top D^\mu \Phi\xi^* - \gamma\Phi^\top D^\mu P^\pi \Phi\xi^* = \Phi^\top D^\mu R^\pi. \tag{5}$$

To simplify the expressions, let use introduce one more piece of notations:

$$A := \mathbb{E}_{s\sim d^{\mu}(s), s'\sim P^{\pi}(s'|s)}[\phi(s)(\phi(s) - \gamma\phi(s'))^{\top}] = \Phi^{\top}D^{\mu}\Phi - \gamma D^{\mu}P^{\pi}\Phi \in \mathbb{R}^{n\times n},$$

$$b := \mathbb{E}_{s\sim d^{\mu}(s), a\sim\pi(a|s), s'\sim P(s'|s,a)}[r(s,a,s')\phi(s)] = \Phi^{\top}D^{\mu}R^{\pi} \in \mathbb{R}^{n\times 1}.$$

Even though we can use arbitrary distribution, for simplicity we assume stationary distribution of $\mu$. Now, we can rewrite (5) compactly as

$$A\xi^* = b. \tag{6}$$

The corresponding O.D.E. for TD-learning can be written as $\dot{\xi}_t = A\xi_t - b, \xi_0 \in \mathbb{R}^n$. Using the coordinate transform $x_k := \xi_k - \xi^*$, we get the O.D.E. $\dot{x}_t = Ax_t, x_0 \in \mathbb{R}^n$, whose origin is globally asymptotically stable equilibrium point if $\rho(s,a) = \frac{\pi(a|s)}{\mu(a|s)} = 1$ for all $(s,a) \in \mathcal{S} \times \mathcal{A}$. Throughout the paper we will use the vector $x_k := \xi_k - \xi^*$ to represent the coordinate transform of $\xi_k$ to the origin, and will use $\xi_t$ and $x_t$ to denote the corresponding continuous-time counterparts of $\xi_k$ and $x_k$, respectively.

## 2.6 GRADIENT TEMPORAL DIFFERENCE LEARNING

To fix the instability issue of off-policy TD-learning under linear function approximation, Sutton et al. (2008) and Sutton et al. (2009) introduced various stable off-policy TD-learning algorithms, called GTD (gradient TD-learning), GTD2, and TDC (temporal difference correction). The idea behind these algorithms is to minimize the mean-square error of projected Bellman equation (MSPBE) $\min_{\xi\in\mathbb{R}^n} \frac{1}{2}||\Phi^{\top}D^{\mu}(R^{\pi}+\gamma P^{\pi}\Phi\xi-\Phi\xi)||^2_{(\Phi^{\top}D^{\mu}\Phi)^{-1}}$, where $||x||_D := \sqrt{x^{\top}Dx}$, and the global minimizer of MSPBE corresponds to the solution of (6). The core idea of the algorithms is to introduce an additional variable $\lambda_k \in \mathbb{R}^n$ to approximate the stochastic gradient descent method for MSPBE as an objective function. In particular, GTD2 update can be written as

$$\lambda_{k+1} = \lambda_k + \alpha_k(-\phi_k^{\top}\lambda_k + \rho_k\delta_k(\xi_k))\phi_k, \quad \xi_{k+1} = \xi_k + \alpha_k(\phi_k^{\top}\lambda_k\phi_k - \rho_k\gamma\phi_k^{\top}\lambda_k\phi_k').$$

We denote $\lambda_t$ to denote continuous time part of $\lambda_k$. Since the fixed point for $\lambda_k$ is zero, it doesn't require coordinate transformation. It is a single time-scale algorithm because it uses a single step-size $\alpha_k$. The corresponding O.D.E. is expressed as $\dot{\lambda}_t = -C\lambda_t - Ax_t, \dot{x}_t = A^{\top}\lambda_t$, where $C := \mathbb{E}_{s\sim d^{\mu}(s)}[\phi(s)\phi^{\top}(s)] = \Phi^{\top}D^{\mu}\Phi \in \mathbb{R}^{n\times n}$. Similarly, TDC update can be written as

$$\lambda_{k+1} = \lambda_k + \alpha_k(-\phi_k^{\top}\lambda_k + \rho_k\delta_k(\xi_k))\phi_k \tag{7}$$

$$\xi_{k+1} = \xi_k + \beta_k(-\rho_k\gamma\phi_k^{\top}\lambda_k\phi_k' + \rho_k\delta_k(\xi_k)\phi_k), \tag{8}$$

where the step-sizes, $\alpha_k$ and $\beta_k$, satisfy $\alpha_k/\beta_k \to 0$ as $k \to \infty$ and the Robbins and Monro step-size condition (Robbins & Monro, 1951) in (33) in Appendix. It is a two time-scale algorithm because it uses two time-steps, $\alpha_k$ and $\beta_k$.

## 3 DESIGNING TD-LEARNING THROUGH BACKSTEPPING

We briefly explain the motivation for our algorithmic development. Borkar and Meyn theorem (Borkar & Meyn, 2000) in Lemma 2.1 is a typical tool to prove convergence of Q-learning (Borkar & Meyn, 2000; Lee & He, 2019) and TD-learning (Sutton et al., 2009; Lee et al., 2021). Most of the previous works on off-policy TD-learning algorithms (e.g., GTD2 and TDC) first start with an objective function, and then derive GTD algorithms based on optimization perspectives. Then, the convergence is proved using the corresponding O.D.E. models and stability theory of linear time-invariant systems. A natural question arises is, can we derive off-policy TD-learning algorithms following a reversed step? In other words, can we develop a stable O.D.E. model first using tools in control theory, and then recover back the corresponding off-policy TD-learning algorithms? In this paper, we reveal that a class of off-policy TD-learning algorithms can be derived based on purely control theoretic motivations following such a reversed process. By doing so, this work provides additional insights on off-policy TD-learning algorithms and gives a sound theoretical foundation on off-policy TD-learning algorithms for further developments of new algorithms.

Designing stabilizing control laws for continuous-time nonlinear system has been successful over the past decades (Khalil, 2015). One such technique, so called backstepping, is a popular controller

design method in non-linear control literature (Khalil, 2015). With the help of the backstepping method (Khalil, 2015), we design stabilizing control laws for continuous-time systems, and then the corresponding off-policy TD-learning algorithms are derived, and are shown to be convergent via Borkar and Meyn theorem (Borkar & Meyn, 2000) in Lemma 2.1. The brief procedure is explained in the following steps: Step 1) Choose an appropriate continuous-time dynamic model such that (a) we can recover the TD-fixed point $\xi^*$ in (6) via its equilibrium point; (b) the corresponding stochastic approximation algorithm can be implementable only through transitions of MDP and accessible data.; Step 2) Using the backstepping method, design a control input to stabilize the dynamic model chosen in Step 1).

## 3.1 BACKSTEPPING TD

Now, we introduce a new off-policy TD-learning algorithm, which we call Backstepping TD (BTD). Firstly, we will develop a stabilizing control law for the following the continuous-time system:

$$\dot{\lambda}_t = (-C + \eta A)\lambda_t - Ax_t \tag{9}$$

$$\dot{x}_t = u_t \tag{10}$$

The idea stems from finding a control system for which we can easily apply the backstepping techinque. In details, the backstepping techinqiue can be applied to the two interconnected systems where one subsystem, namely (4), can be stabilized with $x_t$ in (4) as a control input. Therefore, our first aim is to find such a system. To this end, we can try a natural choice of O.D.E. to solve the TD problem, i.e., $\dot{\lambda}_t = A\lambda_t$, which is however unstable in the off-policy case. Therefore, we can develop a modified O.D.E. $\dot{\lambda}_t = (-C + \eta A)\lambda_t - Ax_t$, where $x_t$ is the control input, the negative definite matrix $-C$ is introduced to stabilize the system, and $\eta > 0$ is introduced to provide additional degrees of freedom in design. Now, the constructed system can be stabilized through the state-feedback controller $x_t = \eta \lambda_t$ and admits the simple control Lypaunov function $V(\lambda) = ||\lambda||^2$. Moreover, $A$ should be included in the right-hand side in order to implement the corresponding algorithm without knowing the solution because $x_k = \xi_k - \xi^*$ and $\xi^*$ should be removed using $A\xi^* = b$ in the final step. Simply setting $x_t = \eta \lambda_t$ may cancel out $A$ in the right-hand side, the O.D.E. becomes $\dot{\lambda}_t = -C\lambda_t$, Therefore, as mentioned before, we can apply the backstepping technique by adding an additional dynamic controller. As the next step, the backstepping technique is applied, and one needs to observe what would be the final form of the control system. In summary, if we consist $f(\lambda_t)$ with the combination of $A$ and $-C$ (not necessarily $-C$, it may be $-I$) , it can be a reasonable candidate to apply the backstepping technique. Cancelling $A$ with virtual input only leaves $-C$, which guarantees stability from its negative definiteness. Therefore, (9) and (10) is a reasonable candidate for the dynamics where we can apply the backstepping technique. In particular, our aim is to design an appropriate control input $u_t$ for the above system such that the origin is the unique asymptotically stable equilibrium point, i.e., $(\lambda_t, x_t) \rightarrow 0$ as $t \rightarrow \infty$ for any $(\lambda_0, x_0) \in \mathbb{R}^n \times \mathbb{R}^n$. The overall procedure is depicted in Figure 1 in the Appendix, and we show how to choose the control input $u_t$ in the following lemma.

**Lemma 3.1.** *Consider the O.D.E. in (9) and (10). If we choose the control input $u_t := (A^\top + \eta^2 A - \eta C)\lambda_t - \eta Ax_t$, then the above O.D.E. has globally asymptotically stable origin, i.e., $(\lambda_t, x_t) \rightarrow (0, 0)$ as $t \rightarrow \infty$ for any $(\lambda_0, x_0) \in \mathbb{R}^n \times \mathbb{R}^n$.*

*Proof sketch.* The proof follows the steps given in the backstepping scheme in Section 3. First, substituting $x_t$ in (9) with a virtual controller $\tilde{x}(\lambda_t)$, we will design a control law $\tilde{x}(\lambda_t)$ that stabilizes the following new virtual system:

$$\dot{\lambda}_t = (-C + \eta A)\lambda_t - A\tilde{x}(\lambda_t). \tag{11}$$

One natural choice of the virtual controller is $\tilde{x}(\lambda_t) = \eta \lambda_t$. Plugging it into (11) leads to $\dot{\lambda}_t = -C\lambda_t$, and we can verify the global asymptotic stability of the above system with the following Lyapunov function:

$$V(\lambda_t) := \frac{||\lambda_t||_2^2}{2}. \tag{12}$$

We now consider the original O.D.E. in (9) and (10). Applying simple algebraic manipulations yield $\dot{\lambda}_t = -C\lambda_t - A(x_t - \eta \lambda_t), \quad \dot{x}_t = u_t$. The error between $x_t$ and the virtual controller $\tilde{x}(\lambda_t)$ can

be expressed as new variable $z_t$, which is $z_t := x_t - \tilde{x}(\lambda_t) = x_t - \eta\lambda_t$. Rewriting the O.D.E. in (9) and (10) with $(\lambda_t, z_t)$ coordinates, we have

$$\dot{\lambda}_t = -C\lambda_t - Az_t \tag{13}$$
$$\dot{z}_t = u_t + \eta C\lambda_t + \eta Az_t.$$

To prove the global asymptotic stability of the above system, consider the function $V_c(\lambda_t, z_t) := V(\lambda_t) + \frac{1}{2}||z_t||_2^2$ where $V(\lambda_t)$ is defined in (12). By taking $u_t$ as $u_t = A^\top\lambda_t - \eta C\lambda_t - \eta Az_t$, we can apply LaSall's invariance principle in Lemma 7.1. The full proof is in Appendix Section 7.4.1. $\quad\square$

Using the relation $z_t := x_t - \eta\lambda_t$, the control input in the original coordinate $(\lambda_t, x_t)$ can be written as $u_t := A^\top\lambda_t - \eta C\lambda_t - \eta Az_t = (A^\top + \eta^2 A - \eta C)\lambda_t - \eta Ax_t$. Plugging this input into the original open-loop system in (9) and (10), the closed-loop system in the original coordinate $(\lambda_t, x_t)$ can written as

$$\dot{\lambda}_t = (-C + \eta A)\lambda_t - Ax_t \tag{14}$$
$$\dot{x}_t = (A^\top + \eta^2 A - \eta C)\lambda_t - \eta Ax_t, \tag{15}$$

whose origin is also globally asymptotically stable according to Lemma 3.1. Recovering back from $x_t$ to $\xi_t$, we have $\frac{d}{dt}\begin{bmatrix}\lambda_t \\ \xi_t\end{bmatrix} = \begin{bmatrix} -C + \eta A & -A \\ A^\top + \eta^2 A - \eta C & -\eta A \end{bmatrix}\begin{bmatrix}\lambda_t \\ \xi_t\end{bmatrix} + \begin{bmatrix} b \\ \eta b \end{bmatrix}$. The corresponding stochastic approximation of the O.D.E. in Theorem 3.1 becomes

$$\lambda_{k+1} = \lambda_k + \alpha_k(((-1+\eta)\phi_k^\top - \eta\rho_k\gamma\phi_k'^\top)\lambda_k + \rho_k\delta_k(\xi_k))\phi_k \tag{16}$$
$$\xi_{k+1} = \xi_k + \alpha_k(((-\eta+\eta^2)\phi_k^\top - \eta^2\rho_k\gamma\phi_k'^\top)\lambda_k\phi_k + \eta\rho_k\delta_k(\xi_k)\phi_k + (\phi_k^\top\lambda_k\phi_k - \rho_k\gamma\phi_k^\top\lambda_k\phi_k')). \tag{17}$$

The equilibrium point of the above O.D.E. is $(0, \xi^*)$. Hence, we only need to transform the coordinate of $\xi_t$ to $x_t = \xi_t - \xi^*$, which results to the O.D.E. in (14) and (15). With the above result, we are now ready to prove convergence of Algorithm 1. The proof simply follows from Borkar and Meyn theorem in Lemma 2.1, of which the details can be found in Sutton et al. (2009).

**Theorem 3.1.** *Under the step size condition (33), with Algorithm 1 in Appendix, $\xi_k \to \xi^*$ as $k \to \infty$ with probability one, where $\xi^*$ is the fixed point of (6).*

*Proof.* The proof is done by checking Assumption 7.1 in Appendix. $\quad\square$

**Remark 3.1.** *Theorem 3.1 doesn't require any condition on $\eta$. Therefore, we can set $\eta = 0$, which results to GTD2 developed in Sutton et al. (2009).*

## 3.2 RECOVERING SINGLE TIME-SCALE TDC

In this section, we derive a single-time scale version of TDC (Sutton et al., 2009) through the back-stepping design in the previous section. TDC (Sutton et al., 2009) was originally developed as a two-time scale algorithm in Sutton et al. (2009). Even though the two time-scale method provides theoretical guarantee for a larger class of algorithms, the single time-scale scheme provides more simplicity in practice, and shows faster convergence empirically. Subsequently, Maei (2011) provided a single-time scale version of TDC by multiplying a large enough constant $\eta > 0$ to the faster time scale part (7), which leads to

$$\lambda_{k+1} = \lambda_k + \beta_k\eta(-\phi_k^\top\lambda_k + \rho_k\delta_k(\xi_k))\phi_k \tag{18}$$
$$\xi_{k+1} = \xi_k + \beta_k(-\rho_k\gamma\phi_k^\top\lambda_k\phi_k' + \rho_k\delta_k(\xi_k)\phi_k), \tag{19}$$

where

$$\eta > \max\left\{0, -\lambda_{\min}\left(C^{-1}(A + A^\top)/2\right)\right\}. \tag{20}$$

Here, we derive another version of single-time TDC by multiplying a constant to the slower time-scale part in (8), which results in

$$\lambda_{k+1} = \lambda_k + \alpha_k(-\phi_k^\top\lambda_k + \rho_k\delta_k(\xi_k))\phi_k \tag{21}$$
$$\xi_{k+1} = \xi_k + \alpha_k\beta(\phi_k^\top\lambda_k\phi_k - \rho_k\gamma\phi_k^\top\lambda_k\phi_k' + \rho_k\delta_k(\xi_k)\phi_k), \tag{22}$$

where $\beta$ satisfies

$$0 < \beta < -\frac{\lambda_{\min}(C)}{\lambda_{\min}(A)} \quad \text{if} \quad \lambda_{\min}(A) < 0, \text{ else} \quad \beta > 0. \tag{23}$$

We can derive the above algorithm following similar steps as in Section 3.1. Let us first consider the following dynamic model:

$$\dot{\lambda}_t = -C\lambda_t - Ax_t \tag{24}$$

$$\dot{x}_t = u_t \tag{25}$$

Using the backstepping technique, we can prove that the above system admits the origin as a global asymptotically stable equilibrium point with the control input $u_t := \beta\left((A^\top - C)\lambda_t - A\xi_t\right)$, which is shown in the following lemma:

**Lemma 3.2.** *Consider the O.D.E. in (24) and (25). Suppose that we choose the control input $u_t := \beta\left((A^\top - C)\lambda_t - A\xi_t\right)$, and $\beta$ satisfies condition (23). Then, the above O.D.E. has globally asymptotically stable origin, i.e., $(\lambda_t, x_t) \to (0,0)$ as $t \to \infty$.*

The proof of Lemma 3.2 is given in Appendix Section 7.4.2. By Borkar and Meyn theorem in Lemma 2.1, we can readily prove the convergence of Algorithm 2 in Appendix, which uses stochastic recursive update (21) and (22).

**Theorem 3.2.** *Consider Algorithm 2 in Appendix. Under the step size condition (33), and if $\beta$ satisfies (23), $\xi_k \to \xi^*$ as $k \to \infty$ with probability one, where $\xi^*$ is the fixed point of (6).*

We will call the Algorithm 4 as TDC-slow, and single-time version of TDC suggested by Maei (2011) as TDC-fast. Other than the multiplication of a constant reflecting two-time scale property, we can make TDC into a single-time algorithm, which we call a single time-scale TDC2, while the original version in Maei (2011) will be called the single time-scale TDC. The derivation is given in Appendix Section 7.5. The performance of such versions of TDC are evaluated in Appendix Section 7.9.1. Even though not one of the algorithms outperforms each other, TDC-slow and TDC2 shows better performance in general.

## 3.3 GENERALIZING TDC++

This section provides versions of TDC++ (Ghiassian et al., 2020), which is variant of TDC. With an additional regularization term $\xi_k$ on both updates of TDC in (7) and (8), the update is written as follows:

$$\lambda_{k+1} = \lambda_k + \alpha_k\eta(-\phi_k^\top\lambda_k + \rho_k\delta_k(\xi_k))\phi_k - \beta\lambda_k) \tag{26}$$

$$\xi_{k+1} = \xi_k + \alpha_k(-\rho_k\gamma\phi_k^\top\lambda_k\phi_k' - \beta\lambda_k + \rho_k\delta_k(\xi_k)\phi_k), \tag{27}$$

where $\eta > 0$ satisfies (20) and $\beta > 0$ is a new parameter. Note that TDC++ can be simply viewed as variant of TDC by adding the term $\beta\lambda_k$ in the update, which can be seen as a regularization term. Therefore, letting $\beta = 0$ yields the original TDC. In this paper, we prove that our controller design leads to the following update:

$$\lambda_{k+1} = \lambda_k + \alpha_k\eta(-\phi_k^\top\lambda_k + \rho_k\delta_k(\xi_k))\phi_k - \beta\lambda_k) \tag{28}$$

$$\xi_{k+1} = \xi_k + \alpha_k(-\rho_k\gamma\phi_k^\top\lambda_k\phi_k' + (1-\kappa\eta)\phi_k^\top\lambda_k\phi_k - \kappa\beta\eta\lambda_k + \rho_k\kappa\eta\delta_k(\xi_k)\phi_k), \tag{29}$$

where $\kappa$ and $\beta$ are new parameters and when $\kappa = 1/\eta$ it becomes TDC++. The difference with the original TDC++ can be seen in their corresponding O.D.E. forms. The corresponding O.D.E. for (26) and (27) (original TDC++) can be expressed as: $\frac{d}{dt}\begin{bmatrix}\lambda_t \\ x_t\end{bmatrix} = \begin{bmatrix} -\eta(C + \beta I) & -\eta A \\ A^\top - C - \beta I & -A \end{bmatrix}\begin{bmatrix}\lambda_t \\ x_t\end{bmatrix}$.

Meanwhile, the O.D.E. corresponding to (28) and (29) (new TDC++) becomes $\frac{d}{dt}\begin{bmatrix}\lambda_t \\ x_t\end{bmatrix} = \begin{bmatrix} -\eta(C + \beta I) & -\eta A \\ A^\top - \kappa\eta(C + \beta I) & -\kappa\eta A \end{bmatrix}\begin{bmatrix}\lambda_t \\ x_t\end{bmatrix}$. We experiment under different of $\kappa$ and $\eta$ to examine the behavior of new TDC++. The result shows that in general, smaller $\kappa$ leads to better performance. The results are given in Appendix Section 7.9.

**Lemma 3.3.** *Consider the following O.D.E.:*

$$\dot{\lambda}_t = -\eta(C + \beta I)\lambda_t - \eta A x_t \tag{30}$$

$$\dot{x}_t = u_t. \tag{31}$$

*Suppose that we choose the control input $u_t := (A^\top - \kappa\eta(C + \beta I))\lambda_t - \kappa\eta A x_t$. Assume $\eta > 0$ and $\beta$ and $\kappa$ satisfies the following condition: $\beta + \kappa\lambda_{\min}(A) > \lambda_{\min}(C)$. Then, the above O.D.E. has globally asymptotically stable origin, i.e., $(\lambda_t, x_t) \to (0, 0)$ as $t \to \infty$.*

The proof is given in Appendix Section 7.4.3. With Lemma 2.1, we can prove the convergence of stochastic update with (28) and (29) whose pseudo code is given in Algorithm 5 in Appendix.

**Theorem 3.3.** *Consider Algorithm 5 in Appendix. Under the step-size condition (33) and if $\eta$ satisfies (20), then $\xi_k \to \xi^*$ as $k \to \infty$ with probability one, where $\xi^*$ is the TD fixed point in (6).*

**Remark 3.2.** *We can replace the regularization term with nonlinear terms satisfying certain conditions. The details are given in Appendix Section 7.6.*

## 4 EXPERIMENTS

We verify the performance and convergence of the proposed BTD under standard benchmarks to evaluate off-policy TD-learning algorithms, including Baird environment (Baird, 1995), RandomWalk (Sutton et al., 2009) with different features, and Boyan chain (Boyan, 2002). The details about the environments are given in Appendix Section 7.7. From the experiments, we see how BTD behaves under different coefficients $\eta \in \{-0.5, -0.25, 0, 0.25, 0.5\}$. We measure the Root Mean-Squared Projected Bellman Error (RMSPBE) as the performance metric, and every results are averaged over 100 runs. From Table 1, the result with $\eta = 0.5$ shows the best performance except at Baird, where $\eta = 0$, corresponding to GTD2 performs best. There exist two aspects on the role of $\eta$. First of all, it can be thought of as a parameter that can mitigate the effect of instability coming from matrix $A$ in (9). For example, a smaller $\eta$ can stabilize the system. However, as a trade off, if $\eta$ is too small, then the update rate might be too small as well. As a result, the overall convergence can be slower. Furthermore, $\eta$ also controls the effect of $-C$ in (13) in the BTD update rules, where $-C$ corresponds to $(-\eta + \eta^2)\phi_k^\top \lambda_k \phi_k$ in (17). Note that the role of $\eta$ in the final BTD update rule in (17) shows different perspectives compared to that in (9). In particular, $\eta = 1/2$ maximizes the effect of $-C$ in (17). From Table 1, it leads to reasonably good performances in most domains. Another natural choice is to multiply $\eta$ to $-C$ instead of $A$. However, in such cases, we need to introduce another constrain $\eta > 0$, whereas in the current BTD, convergence is guaranteed for all $\eta \in \mathbb{R}$. Finally, we note that simply multiplying $-C$ by a large positive constant does not lead to good results in general. This is because in this case, it may increase variance, and destabilize the algorithm. Overall results are given in Appendix Section 7.8.

Table 1: Backstepping TD, step-size = 0.01

| $\eta$ \\ Env | -0.5 | -0.25 | 0 | 0.25 | 0.5 |
|---|---|---|---|---|---|
| Boyan | $1.51 \pm 0.66$ | $1.481 \pm 0.656$ | $1.452 \pm 0.647$ | $1.428 \pm 0.64$ | $1.408 \pm 0.635$ |
| Dependent | $0.11 \pm 0.19$ | $0.097 \pm 0.163$ | $0.086 \pm 0.142$ | $0.079 \pm 0.128$ | $0.076 \pm 0.122$ |
| Inverted | $0.21 \pm 0.25$ | $0.173 \pm 0.218$ | $0.151 \pm 0.193$ | $0.139 \pm 0.177$ | $0.136 \pm 0.172$ |
| Tabular | $0.17 \pm 0.28$ | $0.147 \pm 0.238$ | $0.133 \pm 0.208$ | $0.124 \pm 0.191$ | $0.122 \pm 0.188$ |
| Baird | $0.1 \pm 0.64$ | $0.09 \pm 0.629$ | $0.085 \pm 0.625$ | $0.087 \pm 0.628$ | $0.092 \pm 0.637$ |

## 5 CONCLUSION

In this work, we have proposed a new framework to design off-policy TD-learning algorithms from control-theoretic view. Future research directions would be extending the framework to non-linear function approximation setting.

## 6 ACKNOWLEDGEMENTS

This work was supported by the National Research Foundation under Grant NRF-2021R1F1A1061613, Institute of Information communications Technology Planning Evaluation (IITP) grant funded by the Korea government (MSIT)(No.2022-0-00469), and the BK21 FOUR from the Ministry of Education (Republic of Korea). (Corresponding author: Donghwan Lee.)

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
