# OpenReview forum: "Backstepping Temporal Difference Learning"
_ICLR.cc/2023/Conference — ICLR 2023 poster_

### Official Review · Reviewer_8KPk · 2022-10-17

**Confidence:** 4
**Clarity, Quality, Novelty And Reproducibility:** quality, clarity and originality are …
**Correctness:** 4
**Technical Novelty And Significance:** 3
**Empirical Novelty And Significance:** 3
**Recommendation:** 6

**Strength And Weaknesses:**

Strength:
This paper is well written and easy to follow. Using back-stepping to develop stable off-policy RL algorithms appears novel to me and can probably open up a new direction in RL. The authors did a good job in connecting their methods with existing ones. Though I didn't check the proof line by line, I feel confident that the proof should hold.

Weaknesses:
My major concern is that the proposed approach feels so hand-crafted.
The development of the new algorithm starts from the ODEs in (9) and (10). I absolutely have no idea where the two ODEs come from. It seems that the ODE in (9) comes from the ODE just above (7), which is the ODE from GTD2 updates. But how do the author motivate the introduction of \eta in (9). Even we can accept that the authors somehow magically introduced this \eta, motivating (9) with the ODE from GTD still does not make sense to me. **If we do not know GTD beforehand, is there still a reasonable way to write down the ODEs in (9) and (10)?** If the authors cannot provide an affirmative and convincing answer, I do not think the proposed framework of using backing stepping is significant -- it can only start from existing solutions and cannot start from the problem

**Summary Of The Paper:**

This paper presents a brand new approach for developing gradient TD methods from canonical control theories. The approach starts with finding an ODE of interest and then uses back-stepping to stabilizing the ODE. The approach succeeded in both recovering existing GTD methods and developing some variants.

**Summary Of The Review:**

See above

---

> ### Author Response · Authors · 2022-11-16
> **Reply to Reviewer 8KPk**
>
> We are grateful to the reviewer for constructive comments on our
> manuscript, which are very helpful in improving the quality of the
> paper. Please note that the changes are marked with blue fonts
> in the revised manuscript.
>
> -   *Origin of equation (9) and (10) :* Please refer to our common
>     response.
>
> -   *Role of $\eta$ :* Please refer to our common response.
>
> -   *Difference between GTD and BTD:* The O.D.E. dynamics of BTD is
>     different from GTD2 and TDC as can be seen in their O.D.E. dynamics.
>     The O.D.E. dynamics of BTD coincides to that of GTD2 when setting
>     $\eta =0$, cancelling out some components in the O.D.E. dynamics of
>     BTD. Hence, BTD can be considered as generalization of GTD2.

---

### Official Review · Reviewer_t2qM · 2022-10-24

**Confidence:** 4
**Correctness:** 3
**Technical Novelty And Significance:** 3
**Empirical Novelty And Significance:** 3
**Recommendation:** 6

**Clarity, Quality, Novelty And Reproducibility:**

=== Clarity ===

The paper is written clearly overall. Presentation-wise, certain equations can be presented in a more visually clear manner. General readers will also appreciate more in-text explanations behind the equations.

=== Quality ===

The paper has relatively high technical quality.

=== Novelty ===

The paper is novel as far as I can see, in that it provides a novel application of non-linear control theory to the TD-learning case and has managed to derive a new algorithm too.

=== Reproduce ===

Experiment results are fairly simple and should be easily reproducible.

**Details Of Ethics Concerns:**

No concerns.

**Strength And Weaknesses:**

=== Strength ===

The paper seems to provide a novel and unifying perspective on viewing various TD-learning algorithms under linear function approximations in the off-policy learning setting. Prior work such as GTD and TDC, which have been motivated to bypass the instability of vanilla TD-learning, can be understood as special instances under the new framework proposed in this paper. The paper also suggests a new stable TD-learning algorithm, which is quite interesting in its own right.

=== Weakness ===

Overall, the intuition of the new algorithm is not very clear, despite the fact that it has been derived based on control Lyapunov function and leads to stable learning, and the fact that it can be implemented as a single time scale algorithm. The paper can be greatly improved if further intuitions and explanations of the new algorithm can be provided, beyond control-theoretic mathematical aspect of the algorithm. The experiment section of the paper is also quite simple, making it more difficult to assess the advantage of the new algorithm.

**Summary Of The Paper:**

The paper studies TD-learning for a non-linear control theory perspective. Motivated by the idea of designing stabilizing control policies for non-linear ODE systems based on the control Lyapunov function method, the paper studies TD-learning as a special case. The paper designs a novel TD-learning algorithm that is stable under linear function approximation in the off-policy learning setting, and provides extensions to GTD2 and TDC. The paper is mostly theoretical and uses a few experiments to illustrate properties of the new algorithm.

**Summary Of The Review:**

A few questions and comments.

=== **Notation** ===

In Eqn 4, I suggest the author used notation other than $\lambda_t$ for the state variable. Maybe $\lambda_t$ is a standard notation in the control literature, since much of the audience has RL background, it might be better to use something like $y_t$ as the state variable, which also distinguishes from the controllable state $x_t$.

The notation $x_s(\lambda_t)$ is confusing. From what I understand, $x_s(\cdot)$ is itself a function and the index $s$ does not stand for anything. Maybe it is better to replace $x_s$ by something like $\tilde{x}$ altogether because it can be easy for certain readers to relate $s$ with the time index $t$, leading to confusions.

=== **TD-learning** ===

I think a factor of $\phi_k$ is missing from the TD-learning update in Sec 2.5, TD-learning should update as
$$e_{k+1}=e_k+\alpha_k\rho_k\delta_k\phi_k$$
where $\phi_k$ is necessary to match the dimension against $e_k\in\mathbb{R}^n$.

In Eqn 5, $X$ should be $\Phi$.

=== **Motivation for Sec 2.6** ===

Presentation-wise, as an example, I think it would be good for the paper to provide more motivations for the behavior of GTD, GTD2 and TDC. These algorithms have been designed to improve stability of vanilla TD-learning -- can we spot from the vanilla TD-learning's update rule that it is unstable, can we say something about the non-PD of the $A$ matrix defined in Eqn 6, can we show some plots displaying the instability? This provides motivations for the class of stable TD-learning algorithms that provide motivation for this work.

I also hope the authors can provide more elaborations on the background in Sec 2.6, where they say "the idea behind these algos is to minimize the mean-square error of MSPBE". I don't immediate see how the idea of minimizing mean-square error of MSPBE is reflected in Eqn 7-8 or equations right above them. Maybe it is good to give a quick explanation here. It is also good to explain why minimizing MSPBE helps improve stability, which is not immediately clear.

=== **Sec 3.1** ===

It is a bit confusing where Eqn 9-10 come from, somehow I feel they are related to the ODE systems on page 9 right above Eqn 7, but no clear explanation is given. Eqn 9-10 defer from the ODE system right above Eqn 7 by a factor of $-\eta A\lambda_t$, whose motivation is not clear to me. The authors should provide a more clear motivation as to how one arrives at such an ODE system to start with and why adding the extra term $-\eta A\lambda_t$.

=== **BTD** ===

BTD in Eqn 14-15 are the main contributions of the paper. Though it is good that the authors have arrived at a novel TD-learning update rule that is stable in the off-policy case, overall it feels like it is not completely clear how one arrives at the new update rule in an "easy-to-understand" way. This can be better strengthened and clarified by providing more motivations for each step of the derivation, such as the origin of the ODEs in Eqn 9-10.

Meanwhile, given Eqn 16-17, it is better if the authors can provide further motivations and intuitions for the updates. Decomposing the terms in the ODE updates, can we deliver interpretation intuitions for each term, do they correspond to certain types of modified TD errors? Since the algorithm is single time-scale, it will be interesting to understand exactly what mechanism leads to the stability of the new update rule, beyond the set of math that arrives at the conclusion that the update is stable.

=== **$\eta$** ===

Thus far to me, the role of $\eta$ is mysterious and not well justified in the paper. In Eqn 9-10, the extra factor introduced $\eta$ into the ODE system. Throughout, it is not clear what role $\eta$ plays in stabilizing the ODE system. It seems that $\eta=0$ recovers GTD2, but what is the intuitive explanation for having $\eta>0$? Is it just a numerical trick, does it correspond to certain interpretable regularization?

=== **Experiments** ===

Experiment results are probably too simplistic in the main paper. I'd appreciate a table or plots of comparison of BTD vs. GTD2, TDC and vanilla linear TD, emphatic TD on tabular domains listed in Table 1.

Since the role of $\eta$ is not clear in the BTD update and is poorly motivated, it is not clear what it means when $\eta>0$ works more poorly than $\eta=0$ (GTD2) on the Baird example. Can we characterize when to use $\eta>0$ and when to set $\eta=0$ in practice, does the theory provide any useful indications on that?

=== **Minor comments** ===

The first sentence below Sec 3.3 does not read well.

---

> ### Author Response · Authors · 2022-11-16
> **Reply to Reviewer t2qM (1/2)**
>
> We are grateful to the reviewer for constructive comments on our
> manuscript, which are very helpful in improving the quality of the
> paper. Please note that the changes are marked with blue fonts
> in the revised manuscript.
>
> -   *Notation :* Following the reviewer's suggestion, the notations
>     indicated by the reviewer has been changed.
>
> -   *TD-learning :* There were typos in the initial submission, which
>     have been corrected in the revised version thanks to the reviewer's
>     comment.
>
> -   *Motivation for Sec 2.6 :* The instability of
>     $A:= \Phi^{\top}D^{\mu}\Phi - \gamma \Phi^{\top }D^{\mu}P^{\pi}\Phi$
>     comes from $\Phi^{\top }D^{\mu}P^{\pi} \Phi$ due to the mismatch
>     between behavior policy $\mu$ and target policy $\pi$. In
>     particular, if we use on-policy learning, i.e., $\mu = \pi$, $A$
>     becomes positive definite (page 206 in \[sutton18\]), while in the
>     off-policy setting, $\mu \neq \pi$, $A$ may not be positive definite
>     due to the indefiniteness of $\Phi^{\top }D^{\mu}P^{\pi} \Phi$. To
>     further see how TD and other class of stable TD-learning algorithm
>     behaves, we have newly added O.D.E. plots of algorithms in Appendix
>     Section 6.10.
>
>     Regarding Section 2.6, global minimizer of MSPBE corresponds to
>     TD-fixed point, which is the solution of equation (5). Additionally,
>     we note that directly minimizing MSPBE using stochastic gradient
>     decent methods is not possible because unbiased stochastic gradient
>     samples of MSPBE are not available in general. In particular, the
>     gradient of MSPBE can be expressed as
>     $\mathbb{E}[(\gamma \phi^{\prime}_k-\phi_k)\phi_k^{\top}]\mathbb{E}[\phi_k\phi_k^{\top}]^{-1}\mathbb{E}[(r_k+\gamma \phi^{\prime\top}_k \xi_k - \phi_k^{\top}\xi_k)\phi_k]$,
>     and due to the inverse operation and double sampling issue
>     (dependency of $\phi^{\prime}_k$ in the first term and last term),
>     stochastic samples would lead to significant biases. Hence, GTD2 and
>     TDC tries to approximate the stochastic gradient of MSPBE. Moreover,
>     we have newly added more details on the relation of minimizing MSPBE
>     and stability in Appendix Section 6.11 in this revision.
>
> -   *Origin of Eqn (9) and (10) and BTD:* Please refer to our common
>     response.
>
> -   *Intuitions behind BTD updates:* GTD algorothms are derived from the
>     MSPBE minimization problem. In particular, GTDs and TDC algorithms
>     are obtained by replacing some nonlinear parts of the gradient of
>     MSPBE with its estimation to get unbiased gradient estimations. On
>     the other hand, the proposed BTD is based on the control theoretic
>     argument, i.e., the backstepping technique, and it is in general
>     hard to provide other intuitive meaning of it outside the control
>     theoretic views. Nevertheless, some intuitive meaning of BTD updates
>     we can think of is as follows. Roughly speaking, basic intuition of
>     BTD is to consider the TD-error as control input to the update of
>     $\lambda_{k+1}$ in equation (16). We want TD-error to cancel out
>     $(-\phi_k + \gamma \phi^{\prime})^{\top}\lambda_k$, which is the
>     unstable part of dynamics of $\lambda_k$. Moreover, if TD-error does
>     not vanish, then $\lambda_{k+1}$ does not vanish as well, and this
>     works as a feedback to $\xi_{k+1}$ in equation (17). The update in
>     equation (17) is the result from recovering original coordinate
>     $x_t$ from the error variable $z_t := x_t - \eta \lambda_t$, which
>     can be also seen in Figure 1 in Appendix Section 6.2. One
>     interesting connection with GTD2 is that substituting $u_t$ in the
>     dynamics of $\dot{z}_t$ of the O.D.E. in (13) with the stabilizing
>     control law, the O.D.E. dynamics of the coordinate transformed
>     $(\lambda_t,z_t)$ coincides with the O.D.E. dynamics of GTD2.

---

> > ### Author Response · Authors · 2022-11-16
> > **Reply to Reviewer t2qM (2/2)**
> >
> > -   *Role of $\eta$ :* Please refer to our common response.
> >
> > -   *Experiments:* In Baird's counter example, GTD2 ($\eta = 0$) shows
> >     faster convergence than the proposed BTD ($\eta > 0$). The
> >     algorithms' performance may depend on the MDP's structure, and it is
> >     in general very hard to theoretically analyze convergence properties
> >     for different classifications of MDPs. However, our rough guess is
> >     that the performance may depend on the fact that Baird counter
> >     example is an overparametrized example, where the overparametrized
> >     example means that the dimension of learnable parameters are larger
> >     than the size of the state space. There exist few analysis of
> >     TD-learning under overparametrized setting under restrictive
> >     assumptions \[thomas22on,xiao21\]. The behavior of GTD2 or TDC under
> >     overparametrization case is largely unexplored area to the authors'
> >     knowledge. Another potential reason may be due to the fact that GTD2
> >     can be considered as a primal-dual saddle-point method \[du19\]. In
> >     particular, we conjecture that Baird counter example may be
> >     well-posed for primal-dual optimization, and hence, it may perform
> >     well for the Baird example. Further theoretical analysis of the
> >     phenomenon can be interesting future topics. Moreover, we have newly
> >     added a table of comparison between the algorithms in Appendix
> >     Section 6.9.3. Even though TD and ETD shows good performance in
> >     several domains, it shows unstable behavior in Baird's counter
> >     example. TDC-fast (equation (18) and (19)) shows better performance
> >     than other algorithms in Baird's counter example, but in other
> >     domains, it shows worse performance than TDC-slow (equation (21) and
> >     (22)) or TDC2 (equation (41) and (42)) as can be seen in Table 7 in
> >     Appendix 6.9.1. Moreover, when $\eta = 0.5$, BTD shows better
> >     performance than GTD2 except at Baird's counter example.
> > \
> > &nbsp;
> > \
> > &nbsp;
> >     \[du19\] Du, Simon S., and Wei Hu. \"Linear convergence of the
> >     primal-dual gradient method for convex-concave saddle point problems
> >     without strong convexity.\" The 22nd International Conference on
> >     Artificial Intelligence and Statistics. PMLR, 2019.
> >
> >     \[sutton18\] Sutton, Richard S., and Andrew G. Barto. Reinforcement
> >     learning: An introduction. MIT press, 2018.
> >
> >     \[thomas22on\] Valentin Thomas, . \"On the role of
> >     overparameterization in off-policy Temporal Difference learning with
> >     linear function approximation.\" Advances in Neural Information
> >     Processing Systems. 2022.
> >
> >
> >     \[xiao21\] Xiao, Chenjun, et al. \"Understanding and leveraging
> >     overparameterization in recursive value estimation.\" International
> >     Conference on Learning Representations. 2021.

---

> > > ### Comment · Reviewer_t2qM · 2022-12-12
> > > **Response to your replies**
> > >
> > > Thank you for the detailed response.
> > >
> > > My technical concerns have been mostly addressed. As with the comments from other reviewers, a major suggestion is that the intuition behind the BTD update should be discussed more. Overall, it feels like the motivation for BTD is mainly technical and does not bear as much of a direct intuition as other methods. I'd suggest more discussion around that for the paper to have more impact.
> > >
> > > I will maintain my current evaluation for the paper.

---

### Official Review · Reviewer_kJdF · 2022-10-25

**Confidence:** 4
**Correctness:** 4
**Technical Novelty And Significance:** 3
**Empirical Novelty And Significance:** 3
**Recommendation:** 8

**Clarity, Quality, Novelty And Reproducibility:**

The paper is generally clear, well-written, and contains some original ideas and algorithms.

**Strength And Weaknesses:**

The paper draws ideas from control theory and presents stable and convergent backstepping TD algorithm for off-policy learning. The idea is interesting and seems effective.

Strengths:
+ Interesting combinations of backstepping control with model-free TD learning
+ Well-established analysis and stability results
+ General enough to accommodate existing TD modifications

Weaknesses:
- Backstepping design and theory is suited for nonlinear control systems, but only linear BTD algorithms are discussed here
- Theoretic analysis (e.g., asymptotic/non-asymptotic convergence) of BTD is lacking
- Numerical convergence comparison against existing TD variants not provided



**Summary Of The Paper:**

This paper provides some temporal-difference (TD) learning algorithms based on the celebrated backstepping technique from control theory for addressing a family of nonlinear systems. The proposed TD algorithms are claimed to be stable and convergent in contrast with the divergent behavior of standard TD algorithms. Several existing TD generalizations have been shown as special cases of the proposed backstepping TD algorithm.

**Summary Of The Review:**

Some detailed comments are listed as follows.
1. In the Abstract, please include theoretical and numerical convergence comparison with existing TD algorithms.
2. How Eq. (5) is obtained?
3. The paper focuses only on second-order linear systems. When the order of the system increases, the backstepping technique will have a complexity explosion problem pertaining to computing the derivatives of the virtual control. So would this have an impact on the computational complexity of the proposed algorithm?
4. The paper contains quite some grammar issues.

---

> ### Author Response · Authors · 2022-11-16
> **Reply to Reviewer kJdF**
>
> We are grateful to the reviewer for constructive comments on our
> manuscript, which are very helpful in improving the quality of the
> paper. Please note that the changes are marked with blue fonts
> in the revised manuscript.
>
> -   *In the Abstract, please include theoretical and numerical
>     convergence comparison with existing TD algorithms:* Following the
>     reviewer's comment, we have added brief comments about theoretical
>     and numerical convergence comparison with other TD-learning
>     algorithms in the Abstract of the revised paper. Moreover, further
>     comparisons are given Appendix Section 6.9.3. Even though the
>     standard TD with linear function approximation demonstrates good
>     performance in several domains, it shows unstable behavior in
>     Baird's counter example. TDC-fast (equation (18) and (19)) shows
>     better performance than other algorithms in Baird's counter example,
>     but in other domains, it shows worse performance than TDC-slow
>     (equation (21) and (22)) or TDC2 (equation (41) and (42)) as can be
>     seen in Table 7 in Appendix 6.9.1. Moreover, when $\eta = 0.5$, BTD
>     shows better performance than GTD2 except at Baird's counter
>     example.
>
> -   *How Eq. (5) is obtained?:* It is the so-called projected Bellman
>     equation. When linear function approximation is used, the true
>     TD-fixed point may not lie on the subspace spanned by the feature
>     vectors. To resolve this issue, one common approach is to consider a
>     weighted Euclidean projection after the Bellman operator. The
>     composition of weighted Euclidean projection and Bellman operator is
>     written as
> \begin{aligned}
>     &     \theta^* = \underbrace{\Phi^{\top}(\Phi^{\top}D\Phi)^{-1}
>          \Phi^{\top} D}_{\text{weighted Euclidean Projection}} (R+ \gamma P^{\pi} \Phi \theta^* )\\
>     \iff  \Phi^{\top}D\Phi \theta^*  =   \Phi D (R+ \gamma P^{\pi} \Phi \theta^*).
>     \end{aligned}
> This equation is identical to equation (5). For more
>     details, please refer to \[Sutton09\].
>
> -   *Higher-order system:* To our understanding, our system given in
>     equation (4) consists of two interconnected ODEs with states
>     $\xi_t,x_t$, and each state is of dimension $n$, i.e.,
>     $\xi_t,x_t \in \mathbb{R}^n$. Another potential direction is to
>     augment the state in equation (5) to an augmented state with integer
>     multiple of $n$ dimensions, and this could lead to new algorithms.
>     This extension is not our main interest, and is left for future
>     works.
>
> -   *Nonlinear TD algorithms:* Development of TD-learning algorithms
>     with nonlinear function approximation is still a challenging open
>     problem. There exist some previous works on the topic, e.g.,
>     \[Maei09\], while they require additional steps such as projection,
>     and additional information such as the Hessian matrix to guarantee
>     convergence. Moreover, to the authors' knowledge, nonlinear TD
>     algorithms that are guaranteed to find a reasonable solution have
>     not been fully investigated so far. Therefore, nonlinear TD could be
>     another independent topic that cannot be covered in our paper, and
>     we only focus on linear approximation case in this paper. However,
>     we believe that our framework can be extended to nonlinear
>     TD-learning algorithms with additional works, and this direction is
>     left as a future topic. Finally, even though linear BTD algorithms
>     are addressed in this paper, we also investigate a specific class of
>     nonlinear regularization terms in Appendix Section 6.6.
>
> -   *Asymptotic/Non-asymptotic convergence:* The proof of asymptotic
>     convergence is given in Theorem 3.1, which uses Borkar and Meyn
>     Theorem and Lemma 3.1. Non-asymptotic analysis is another
>     independent and challenging topic, which is not our main focus of
>     this paper. We leave this topic as future works.
>
> -   *Grammar issues:* We have carefully proofread the paper, and
>     corrected grammar and typos comprehensively. Thank you for pointing
>     out the issue.
> \
> &nbsp;
> \
> &nbsp;
>
>     \[Sutton09\] Sutton, Richard S., et al. \"Fast gradient-descent
>     methods for temporal-difference learning with linear function
>     approximation.\" Proceedings of the 26th annual international
>     conference on machine learning. 2009.
>
>     \[Maei09\] Maei, Hamid, et al. \"Convergent temporal-difference
>     learning with arbitrary smooth function approximation.\" Advances in
>     neural information processing systems 22 (2009).

---

### Author Response · Authors · 2022-11-16
**Common Response**

-   We are grateful to the reviewers for their constructive comments on
    our manuscript, which are very helpful in improving the quality of
    the paper. In the following, we list common changes we have made in
    response to these comments, and answer common questions. Please note that the changes are marked
    with blue fonts in the revised manuscript.

-   *Origin of Eqn (9) and (10) and BTD:* Roughly speaking, the idea
    stems from finding a control system for which we can easily apply
    the backstepping technique. In particular, the backstepping
    technique can be applied to two interconnected systems, where one
    subsystem, namely equation (4), can be stabilized with $x_t$ in
    equation (4) as a control input. Therefore, our first aim is to find
    such a system. To this end, we can try a natural choice of O.D.E. to
    solve the TD problem, i.e., $\dot \lambda_t = A \lambda_t$, which is
    however unstable in the off-policy case. Therefore, we can develop a
    modified O.D.E.
    $\dot \lambda_t = (- C +  \eta A) \lambda_t - A x_t$, where $x_t$ is
    the control input, the negative definite matrix $-C$ is introduced
    to stabilize the system, and $\eta >0$ is introduced to provide
    additional degrees of freedom in design. Now, the constructed system
    can be stabilized through the state-feedback controller
    $x_t = \eta \lambda_t$ and admits the simple control Lypaunov
    function $V(\lambda) = ||\lambda ||^2$. Moreover, $A$ should be
    included in the right-hand side in order to implement the
    corresponding algorithm without knowing the solution because
    $x _k = \xi _k  - \xi ^*$ and $\xi ^*$ should be removed using
    $A\xi ^*  = b$ in the final step. Simply setting
    $x_t = \eta \lambda_t$ may cancel out $A$ in the right-hand side,
    the O.D.E. becomes $\dot{\lambda}_t = - C\lambda_t$. Therefore, as
    mentioned before, we can apply the backstepping technique by adding
    an additional dynamic controller. As the next step, the backstepping
    technique is applied, and one needs to observe what would be the
    final form of the control system. In this step, we agree that some
    heuristic trial and error are partially involved, and the BTD
    formulation can be potentially seen as some kinds of discovery
    rather than development. Nevertheless, we would like to stress that
    the above frameworks have been our underlying ideas in developing
    the initial system.

    In summary, if we consist $f(\lambda_t)$ with a linear combination
    of $A$ and $-C$ (not necessarily $-C$, it may be $-I$), it would be
    a reasonable candidate to apply the backstepping technique.
    Cancelling $A$ with virtual input only leaves $-C$, which guarantees
    stability from its negative definiteness. We have newly added the
    above discussions briefly in the revised version.

-   *Role of $\eta$ :* There exist two aspects on the role of $\eta$.

    1.  First of all, it can be thought of as a parameter that can
        mitigate the effect of instability coming from matrix $A$ in the
        initial equation (9). For example, a smaller $\eta$ can
        stabilize the system. However, as a trade off, if $\eta$ is too
        small, then the update rate might be too small as well. As a
        result, the overall convergence can be slower.

    2.  Furthermore, $\eta$ also controls the effect of $-C$ in
        equation (17) in the BTD update rules, where $-C$ corresponds to
        $(-\eta+\eta^2)\phi_k^{\top}\lambda_k\phi_k$ in equation (17).
        Note that the role of $\eta$ in the final BTD update rule
        in (17) shows different perspectives compared to that in (9). In
        particular, $\eta = 1/2$ maximizes the effect of $-C$ in (17).
        As can be seen in Table 1, it leads to reasonably good
        performances in most domains.

    Another natural choice is to multiply $\eta$ to $-C$ instead of $A$.
    However, in such cases, we need to introduce another constrain
    $\eta>0$, whereas in the current BTD, convergence is guaranteed for
    all $\eta \in \mathbb{R}$.

    Finally, we note that simply multiplying $- C$ by a large positive
    constant does not lead to good results in general. This is because
    in this case, it may increase variance, and destabilize the
    algorithm. We have newly added the above discussions briefly in the
    revised version.

---

### Decision · Program_Chairs · 2023-01-20

**Decision:**

Accept: poster

**Justification For Why Not Higher Score:**

As discussed above, the paper offers new perspectives which *could* prove quite useful. To my reading, spotlight presentations should show evidence of having more substantially moved the state-of-the-art.

**Justification For Why Not Lower Score:**

The paper appears to be sound and to tackle a foundational question in RL. New perspectives, especially those that offer new links to a large literature in another field, have the potential to inspire many important future developments.

**Metareview: Summary, Strengths And Weaknesses:**

Temporal difference learning is a foundational idea in reinforcement learning. But the convergence properties of the algorithm have been somewhat mysterious since its introduction. Tsititsiklis and Van Roy (1997) offered some clarity, showing that TD converges with 'on policy' sampling when linear function approximation is used. Counterexamples show divergence is possible with off policy sampling or nonlinear function approximation. Gradient temporal difference learning algorithms [Sutton et. al, 2008] represent another breakthrough, as they provably converge with off policy sampling. In all cases, the problem reduces to understanding whether the solution to a certain ODE is globally asymptotically stable.

This paper uses the backstepping technique from nonlinear control theory to stabilize ODEs arising from the dynamics of TD. This perspective offers a new way to derive existing gradient TD algorithms and lets the authors propose some convergent alternatives.

*Strengths:* The paper offers a fresh perspective that connects two large bodies of academic literature. All revierwers feel the paper is sound and that this connection *could* lead to useful results in the future.

*Weaknesses*: It currently feels like GTD can be understood after-the-fact through the lens of backstepping. But currently, it still seems difficult to derive truly new algorithms from this perspective. It is possible, however, that experts in nonlinear control theory have trained intuition and will find this framework to be a key tool.

**Note From Pc:**

if the above contains the word "oral" or "spotlight" please see: "oral" presentation means -> notable-top-5% and "spotlight" means -> notable-top-25%. As stated in our emails, we are disassociating presentation type from AC recommendations

**Summary Of Ac-Reviewer Meeting:**

We discussed the strengths and weaknesses above. The meeting was helpful in thinking through those. There was universal agreement among the reviewers that this paper is sound and worth accepting.

The review team felt the backstepping technique was a bit unintuitive. Though they could follow the symbolic manipulation, it would be difficult for them to derive new algorithms this way. It is possible that those with more familiarity with nonlinear control theory will feel differently, however.